# Approaches to Pediatric Chest Pain: A Narrative Review

**DOI:** 10.3390/jcm13226659

**Published:** 2024-11-06

**Authors:** Federica Fogliazza, Marina Cifaldi, Giulia Antoniol, Nicolò Canducci, Susanna Esposito

**Affiliations:** Pediatric Clinic, Department of Medicine and Surgery, University of Parma, Via Gramsci, 14, 43125 Parma, Italy; federica.fogliazza95@gmail.com (F.F.); marinacifaldi@icloud.com (M.C.); giulia.antoniol@gmail.com (G.A.); nik.candu92@gmail.com (N.C.)

**Keywords:** chest pain, cardiac evaluation, pediatric emergency, electrocardiograms, chest radiography, echocardiography

## Abstract

Chest pain in children and adolescents is a common reason for emergency department visits and referrals to pediatric cardiologists, often driven by parental concern about potential cardiac causes. However, the vast majority of pediatric chest pain cases are benign and non-cardiac in origin. This narrative review examines the etiology, evaluation, and management of pediatric chest pain, emphasizing the importance of a thorough clinical history and physical examination in distinguishing between benign and serious conditions. This review also explores the role of diagnostic tests such as electrocardiograms, chest radiography, and echocardiography, highlighting the need to balance the avoidance of unnecessary tests with the imperative to rule out life-threatening cardiac conditions. Despite the low prevalence of cardiac causes, the variability in diagnostic approaches underscores the need for standardized evaluation algorithms. These could streamline care, reduce unnecessary resource utilization, and minimize anxiety for both patients and their families. Future studies should focus on assessing the effectiveness of such algorithms in improving clinical outcomes and resource management. The findings underscore the importance of a careful, evidence-based approach to the management of pediatric chest pain.

## 1. Introduction

Chest pain is a frequent reason for pediatric emergency department (ED) visits, often causing significant anxiety for both patients and their families. While the symptom is typically benign, self-limiting, and rarely of cardiac origin in children, its presentation necessitates careful evaluation due to its association with serious conditions in adults. Despite its common occurrence, the exact prevalence of pediatric chest pain remains uncertain, with estimates suggesting it accounts for 0.5–1% of all pediatric ED visits, according to recent literature surveys [1,2,3,4].

In children, the etiology of chest pain can be diverse, encompassing both cardiac and non-cardiac causes. Non-cardiac causes are often psychogenic, musculoskeletal, respiratory, or gastrointestinal in nature. However, in some instances, the precise cause may remain unidentified, leading to a diagnosis of idiopathic chest pain [5,6,7,8]. Given the broad differential diagnosis, healthcare providers must employ a systematic approach to assess chest pain, beginning with a detailed medical history and physical examination to exclude potentially serious underlying conditions [9,10].

Several studies have sought to improve the assessment and management of pediatric chest pain by developing standardized protocols. For instance, Collins et al. identified specific “red flag” features that could indicate a cardiac origin or necessitate further investigation, such as chest X-rays, electrocardiograms (ECGs), echocardiograms, myocardial enzyme testing, or trials of anti-reflux medication and pH impedance monitoring [9,11]. In 2010, Kane et al. introduced the standardized clinical assessment and management plans (SCAMPs) for chest pain, aimed at reducing practice variation, minimizing unnecessary resource utilization, and enhancing clinical outcomes [12]. These protocols are particularly valuable for identifying serious causes of chest pain, including anomalous coronary origins, cardiomyopathy, pulmonary hypertension, myocarditis, pericarditis, aortic dissection, and Takayasu arteritis [13].

The COVID-19 pandemic has introduced additional complexities in the management of pediatric chest pain. As reported by Lubrano et al. [3], the number of ED visits for chest pain decreased during the pandemic. This underscores the need to consider new differential diagnoses, including myocarditis or pericarditis related to SARS-CoV-2 infection or COVID-19 mRNA vaccines [14].

This narrative review aims to synthesize the latest publications to establish a comprehensive framework for the management of pediatric chest pain in the emergency department. By analyzing the most common and possible causes of chest pain in previously healthy children and adolescents (age range 1 months–18 years), we seek to develop a flowchart that can aid pediatricians in distinguishing patients with potentially serious and life-threatening conditions from those with benign causes. This framework will guide clinicians in selecting the most appropriate diagnostic tests, ultimately improving patient outcomes. To achieve this, we conducted a thorough review of studies published in the PubMed database, focusing on articles from the past 15 years that were published in English and relevant to the keywords “chest pain” AND “children” OR “pediatric”.

## 2. First Evaluation

The initial evaluation of chest pain in pediatric patients begins with a thorough history-taking and a comprehensive physical examination. These steps are essential for understanding the characteristics of the pain, such as its quality, duration, location, triggers, and alleviating factors. Identifying alarm signs, such as chest pain associated with exertion, syncope, palpitations, or a family history of cardiac disease, is crucial in assessing the potential severity of the condition [15].

### 2.1. Clinical History

Obtaining a detailed clinical history is a cornerstone in accurately diagnosing the cause of pediatric chest pain and minimizing unnecessary tests. Key elements of the history include the timing and circumstances of chest pain onset [15]. The pain may have started in the morning, before bedtime, or at another specific time. Additionally, understanding the circumstances is equally important, as the pain could have occurred after using the restroom, during exercise, or under other conditions, which can help guide diagnosis. Additional factors to explore include the duration, location, reproducibility (considering exertional or positional factors), and quality of the pain (e.g., crushing central chest pain or pain that worsens with inspiration) [15]. Other important aspects include the radiation of pain, aggravating and relieving factors, associated symptoms (such as dizziness, shortness of breath, or palpitations), recent injuries, febrile infections (e.g., enterovirus, coxsackie virus, or Bornholm’s disease), vaccinations (e.g., myopericarditis secondary to COVID-19 mRNA vaccination and, rarely, to other vaccines), and underlying conditions like connective tissue disorders, asthma, sickle cell disease, or diabetes [15]. Potential thrombophilia, previous thrombosis, smoking habits, and the use of hormonal contraception in female patients are additional risk factors [15]. It is also crucial to inquire about sports activities, any limitations in physical performance, unilateral exertion, and any new or unusual physical strain [15]. A thorough family history, especially of serious cardiac or pulmonary conditions (e.g., arrhythmias, cardiomyopathy, pulmonary hypertension, or sudden death), as well as the patient’s medical and surgical history, particularly previous cardiac history or surgery, should also be explored [15].

### 2.2. Physical Examination

It is essential to assess the patient’s blood pressure, oxygen saturation, heart rate, and respiratory rate, while taking into account the normal physiological variations based on age [16]. Great consideration should be given to age-specific symptoms and causes, as certain conditions are more prevalent in specific age groups [16]. For instance, young children may describe “chest pain” through symptoms like tachycardia or even present with abdominal pain, making it crucial to conduct a comprehensive, full-body examination. This ensures that underlying conditions are not overlooked, as young children may not accurately localize or describe their discomfort. Moreover, in childhood, it is important to focus on growth disturbances, such as whether the patient is short or tall, underweight or obese [16]. Also, the presence or absence of scoliosis and a variety of bone diseases should be noted [16].

Following a comprehensive history, a detailed physical examination can help narrow down the differential diagnosis. For instance, tenderness to palpation and well-localized pain might suggest a musculoskeletal cause, while bruising could indicate trauma, warranting further investigation into potential intrathoracic or extrathoracic injuries. Abnormal lung auscultation could point to a pulmonary origin, and any abnormal cardiac sounds should prompt further assessment for cardiac involvement. If the physical examination findings are unremarkable, a psychogenic cause may be considered, though organic etiologies should not be prematurely excluded [16].

#### 2.2.1. Electrocardiogram (ECG)

An electrocardiogram (ECG) is a quick, noninvasive test that can detect critical cardiac conditions such as acute myocardial infarction, myocarditis, and potentially fatal arrhythmias [7]. Given its utility in identifying serious cardiac issues, an ECG is a fundamental component of the initial evaluation of pediatric chest pain.

#### 2.2.2. Chest Radiography

Chest radiography is a common diagnostic tool used in the evaluation of acute chest pain in children and adolescents. It can reveal conditions such as pneumonia, bronchitis, pneumomediastinum, pneumothorax, pneumopericardium, and cardiomegaly. However, its sensitivity ranges from 11.0% to 17.2% [2,7,17], making it less reliable as a sole diagnostic tool. Despite its limitations, chest radiography is particularly useful for identifying pneumothorax and pneumomediastinum, which, although only accounting for about 3% of pediatric chest pain cases, are critical diagnoses that emergency physicians aim to rule out [4]. Pneumonia, another common pulmonary condition presenting with chest pain, is identified in 3.7–9.3% of pediatric chest pain cases [18]. While chest radiography can identify several diseases, its low sensitivity underscores the importance of correlating radiographic findings with the patient’s clinical history and physical examination [18].

Moreover, the cardiothoracic ratio (CTR) can be a valuable tool in the evaluation of chest pain in children, as it helps to assess whether the heart is abnormally enlarged, which could indicate underlying cardiac pathology [18]. In infants and young children, a CTR of 0.55 to 0.60 is considered normal, while in older children and adolescents, a normal CTR is generally between 0.42 and 0.50. If the CTR exceeds these age-specific thresholds, it may suggest conditions such as cardiomegaly, pericardial effusion, or other heart-related abnormalities that could be causing the chest pain [18]. Thus, measuring the CTR helps differentiate between cardiac and non-cardiac causes of chest pain and is particularly useful in guiding further diagnostic testing and management.

#### 2.2.3. Echocardiogram

An echocardiogram can be used to further evaluate chest pain of suspected cardiac origin by assessing ventricular systolic function, pericardial effusion, and anatomical abnormalities. However, because it requires specialized training to perform and interpret, echocardiography is not routinely used in the pediatric emergency department. Studies show that only 1–1.5% of pediatric patients with acute chest pain undergo echocardiography in the ED [19,20,21]. While there is no universal guideline for when to order an echocardiogram, cardiac point-of-care ultrasound (POCUS) can be a timely and effective tool for assessing global systolic function and pericardial effusion, complementing the echocardiogram in selected cases [18].

#### 2.2.4. Cardiac Troponin

Cardiac troponin is a highly specific and sensitive biomarker for myocardial injury. In adults, it is commonly used to diagnose acute myocardial infarction, and in pediatric patients, it helps identify cardiac conditions such as myocarditis and pericarditis. Brown et al. found that 17.5% of pediatric patients tested for troponin in the ED had elevated levels, with nearly half of these cases ultimately diagnosed as myocarditis or pericarditis [22,23]. However, non-cardiac conditions such as drug intoxication, carbon monoxide poisoning, bronchopneumonia, asthma, shock, sepsis, status epilepticus, and asphyxia can also elevate troponin levels in children. Thus, while troponin is not ideal for screening all pediatric chest pain cases, it is valuable in patients where there is a strong suspicion of cardiac involvement, especially when used alongside a thorough clinical evaluation, ECG, echocardiography, and chest X-ray [23,24,25,26]. Troponin testing in children typically involves measuring either troponin T (TnT) or troponin I (TnI), which are both markers of cardiac muscle injury. TnT is associated with tropomyosin, while TnI reflects actomyosin interactions. Both troponins are highly specific for myocardial damage, but it is not generally necessary to examine them separately, as either marker is sufficient for identifying cardiac injury [23,24,25,26]. Most clinical settings prefer to measure one, often based on local protocols or available assays, since both provide comparable diagnostic information regarding myocardial injury.

Other cardiac enzymes, such as creatine kinase and natriuretic peptide, which are frequently measured in children presenting with chest pain, are generally unnecessary in most cases. These tests do not provide additional diagnostic value in the absence of clinical signs or symptoms suggesting myocardial damage, and their routine use may lead to unnecessary anxiety and resource utilization [23,24,25,26,27,28].

#### 2.2.5. Additional Examinations

Additional tests, such as blood and urine analyses, may be warranted in selected pediatric patients with chest pain. Studies report that 5.6–26.5% of children and adolescents undergo these tests in the ED, with 9–19.2% revealing abnormalities [2,7,19]. Although these tests (i.e., white blood cell count, C-reactive protein, lactate dehydrogenase, and aspartate aminotransferase) are not routinely necessary, they can be crucial in specific clinical contexts. For instance, Drossner et al. found that laboratory tests were more frequently used in patients with cardiac-related chest pain than in those with non-cardiac chest pain (79% vs. 26%) [7].

Cardiac magnetic resonance imaging (MRI) should be considered in children with chest pain when there is a strong suspicion of underlying cardiac pathology that is not fully explained by initial evaluations, such as echocardiography or ECG [1,4,8]. It is particularly useful in cases where myocarditis or pericarditis is suspected, especially following viral infections or post-vaccination, as cardiac MRI is highly sensitive for detecting inflammation of the heart muscle or pericardium. Additionally, if congenital coronary artery anomalies, such as an anomalous origin of the coronary arteries, are suspected but not adequately visualized on other imaging studies, cardiac MRI provides detailed anatomical and functional information. Cardiac MRI is also valuable in diagnosing and evaluating cardiomyopathies, such as hypertrophic or dilated cardiomyopathy, which may be associated with chest pain [29,30,31,32]. In cases where a child has persistent or recurrent chest pain, and initial tests like ECG or echocardiography reveal abnormalities, cardiac MRI can offer a more comprehensive assessment of cardiac structure, function, and tissue characteristics. Furthermore, if a cardiac mass or tumor is suspected based on initial imaging, cardiac MRI can provide further characterization [1,4].

## 3. Red Flags in Pediatric Chest Pain

When evaluating chest pain in children, it is crucial to recognize signs and symptoms that may indicate potentially serious conditions [15,27]. The main “red flags” are listed in Table 1. Recognizing these “red flags” promptly is essential to ensure timely diagnosis and appropriate treatment, preventing potentially serious complications in children with chest pain.

### 3.1. Cardiac Causes

Chest pain has traditionally been viewed as a potentially life-threatening symptom, prompting urgent evaluation and often referral to a pediatric cardiologist, emergency department, or pediatric clinic. Unlike in adults, where chest pain frequently indicates serious cardiac conditions, the majority of chest pain cases in children are benign, with cardiogenic causes being relatively rare [28]. However, the concern that chest pain might signify a cardiac problem drives many parents to seek immediate medical attention, leading to an increased use of healthcare resources, unnecessary diagnostic tests, and heightened anxiety for both the child and their family. The challenge for clinicians is to accurately distinguish between benign chest pain and potentially serious cardiac conditions, ensuring the appropriate use of healthcare resources while minimizing the risk of missing a significant diagnosis.

The “Standardized Clinical Assessment and Management Plans (SCAMPS)” algorithm has been developed to help clinicians strike a balance between careful resource management and the accurate identification of cardiac conditions [13]. A comprehensive evaluation of pediatric chest pain should include a detailed clinical history and physical examination, considering factors such as the duration and location of the pain, whether movement exacerbates the symptoms, the child’s medical and family history (especially regarding cardiac conditions, arrhythmias, congenital heart defects, and sudden death), and findings from the physical examination (including blood pressure, heart rate, auscultation, and palpation of the spine, ribs, and sternum). Based on these findings, appropriate diagnostic tests can be selected to identify or rule out a cardiac cause, thereby conserving clinical resources and reducing the financial burden on families.

Certain characteristics of chest pain in children may suggest a cardiac origin, such as pain that worsens with exercise and improves with rest, associated symptoms like dyspnea, fatigue, or chest pressure, a history of congenital or childhood heart problems, a family history of sudden death or early cardiac disease, abnormal physical examination findings, or an abnormal ECG result [29]. When a cardiac cause is suspected, referral to a pediatric cardiologist is essential [29].

The pediatric cardiologist typically performs a thorough history and physical examination, followed by an ECG and, if indicated, an echocardiogram. In some cases, additional imaging, such as chest radiography or a lung CT scan, may be necessary [30]. Possible cardiac causes of chest pain in children include myocarditis (inflammation of the heart muscle), arrhythmias (abnormal heart rhythms), left ventricular outflow tract obstructions (aortic valve stenosis or coronary artery anomalies), and cardiomyopathies (conditions affecting the size and function of the heart) [29]. Although rare, it is also important to mention congenital anomalies in the course of coronary arteries, such as intramural or interarterial pathways, which can cause chest pain during exercise [29]. These anomalies can be life-threatening if not promptly diagnosed and managed, particularly in physically active children. Additionally, aortic dissection, though uncommon in pediatrics, is a serious cause of severe chest pain, often radiating to the back [29]. This diagnosis must be considered, especially in patients with connective tissue diseases like Marfan syndrome, as it is a medical emergency. Immediate diagnostic and therapeutic intervention is critical to prevent catastrophic outcomes. Recognizing and addressing these conditions early can be life-saving [29,30]. Chest pain is generally not a common symptom of congenital heart disease. Exceptions to this are left ventricular outflow tract obstructions such as aortic valve stenosis or coronary anomalies.

A study conducted at Health World Hospital in Durgapur, West Bengal, India, involving children aged 5–15 years presenting to the emergency department with chest pain, found that only 1 out of 55 patients had a cardiac cause, which was an anomalous origin of the right coronary artery diagnosed through echocardiography and confirmed by CT angiography [1]. This underscores the rarity of cardiac causes in pediatric chest pain cases.

Echocardiography is a widely used, noninvasive, and cost-effective tool in cardiovascular medicine, providing detailed information about the heart’s anatomy, function, and hemodynamics. It is particularly useful for evaluating conditions such as cardiomyopathy, pericardial effusion, valvular heart disease, congestive heart failure, cardiac malformations, and bacterial endocarditis. In emergency settings, echocardiography can also provide critical information for managing conditions like heart failure, acute coronary syndrome, pulmonary embolism, myocardial infarction, sepsis, and endocarditis. Given its accessibility, noninvasiveness, and cost-effectiveness, some researchers recommend echocardiography for all pediatric patients presenting with chest pain [32,33].

However, the necessity of routine echocardiography in pediatric chest pain is debated. Some studies suggest that its diagnostic yield is low in this population, often resulting in normal findings that do not alter clinical management, thus contributing to unnecessary healthcare costs [34]. Therefore, the use of echocardiography should be carefully considered, reserving it for cases where there is a strong clinical suspicion of cardiac involvement.

In conclusion, while idiopathic chest pain is the most common diagnosis in children presenting to pediatric cardiology clinics, cardiac causes remain a rare but important consideration. Echocardiography, while valuable in specific contexts, has limited sensitivity and positive predictive value in this setting and should be used judiciously to avoid unnecessary costs and interventions [21].

### 3.2. Respiratory Causes

Respiratory causes of chest pain in children are relatively rare and are typically diagnosed when associated respiratory symptoms such as cough, wheezing, or radiographic evidence of bronchitis or pneumonia are present [3]. Pleuritis, particularly epidemic pleurodynia, causes severe chest pain and should be excluded [3]. The most common respiratory conditions presenting with chest pain include spontaneous pneumothorax, spontaneous pneumomediastinum, and asthma exacerbations [3].

Spontaneous pneumothorax, although uncommon in children, can be a significant cause of chest pain and respiratory distress. Unlike adults, where lung ultrasound protocols are well established for evaluating dyspnea in the emergency department, there are no specific guidelines for the use of lung ultrasound in pediatric patients. A study conducted at the Bambino Gesù Children’s Hospital in Rome found that lung ultrasound, followed by chest X-ray, effectively diagnosed pneumothorax in children with acute chest pain, demonstrating high sensitivity and specificity [35]. The routine use of lung ultrasound in pediatric emergency settings could reduce the need for chest X-rays, thereby lowering radiation exposure and costs, though further studies are needed to confirm these findings.

Spontaneous pneumomediastinum, characterized by the presence of air in the mediastinum without preceding trauma or medical intervention, is another respiratory condition that can cause chest pain. Esophageal rupture is one of the most common causes for pneumomediastinum [36,37]. It is often triggered by activities that increase intrathoracic pressure, such as coughing, vomiting, or strenuous exercise. A retrospective study of pediatric patients presenting with spontaneous pneumomediastinum found that chest pain, cough, and dysphagia were the most common symptoms, with adolescent males being the most affected group. Diagnosis is typically made via chest radiography, with additional imaging like chest CT being reserved for more complex cases or when further clarification is needed [36].

### 3.3. Musculoskeletal Causes

Musculoskeletal causes are the most common source of non-cardiac chest pain in children, with an incidence ranging from 50% to 68% [37]. This type of pain often results from muscle strain, trauma, or conditions affecting the chest wall. For example, precordial catch syndrome is a benign and often underdiagnosed condition that affects children and adolescents, typically presenting as sudden, sharp chest pain [38]. The pain is often localized to a small area near the heart, usually described over an intercostal space, and is exacerbated by breathing or movement, though it resolves spontaneously within a few minutes without intervention. It is non-life-threatening and does not typically indicate any underlying cardiovascular issues. Studies emphasize the importance of taking a careful patient history to identify the distinct characteristics of the syndrome, which helps differentiate it from more serious conditions such as cardiac disease [39]. Though it often causes significant anxiety in both children and their parents, precordial catch syndrome does not usually require any diagnostic tests, and reassurance is the primary form of treatment [40]. A study in elite swimmers also highlighted the association of precordial catch syndrome with physical exertion, but again emphasized the benign nature of the syndrome even in athletes with asthma [41]. While the exact cause remains unclear, precordial catch syndrome is thought to be related to the irritation of the nerves in the chest wall, particularly in growing children.

Slipping rib syndrome, caused by the irritation of intercostal nerves when a lower rib slips under an adjacent rib, can lead to chest or upper abdominal pain. The diagnosis is confirmed through the “hooking” maneuver, which reproduces the pain and may produce a clicking sensation [30].

Costochondritis, another common musculoskeletal cause, results in sharp, anterior chest pain over multiple costochondral or costosternal junctions, while Tietze syndrome involves localized swelling and pain, typically at a single costosternal or costochondral junction [42]. Additionally, skin conditions like herpes zoster and breast-related conditions (more common in adolescent girls) can contribute to chest wall pain [30].

Postural disorders are also frequently associated with chest pain in children, particularly in cases where abnormalities in spine alignment or muscle tension and flexibility are present [43].

### 3.4. Gastrointestinal Causes

Gastrointestinal conditions account for approximately 2% to 8% of pediatric chest pain cases [44]. Common causes include gastroesophageal reflux disease (GERD) and esophagitis, which can produce retrosternal chest pain that may worsen after eating or when lying down. Other potential gastrointestinal causes include esophageal foreign bodies, caustic ingestion, cholecystitis, and constipation [44].

### 3.5. Psychogenic Causes

Psychogenic factors are responsible for 10% to 30% of chest pain cases, particularly in adolescents [45]. Non-cardiac chest pain in this context may be a manifestation of psychological distress, such as anxiety, depression, or the impact of significant life events. In some cases, the chest pain is associated with hyperventilation syndrome during anxiety attacks. A thorough history is essential to uncover any underlying psychosocial stressors that may be contributing to the pain.

### 3.6. Idiopathic Causes

When no clear organic or psychogenic cause is identified, the chest pain may be classified as idiopathic. It is important to note that some cases of idiopathic chest pain may have an underlying psychopathological basis that remains undetected [2].

## 4. Practical Approach to Pediatric Chest Pain

Chest pain in children and adolescents frequently prompts visits to the emergency department or a pediatric cardiologist, driven largely by parental concern. The causes of pediatric chest pain are diverse, with the vast majority being non-cardiac in origin, as evidenced by numerous studies. However, the fear of missing a potentially life-threatening cardiac condition often leads pediatricians to conduct extensive evaluations. This approach, while well intentioned, can result in unnecessary investigations, medical visits, and hospitalizations, which are both costly and often unwarranted.

A structured approach is essential to avoid the overuse of medical resources while ensuring that serious conditions are not overlooked. Figure 1 illustrates the most common cardiac and non-cardiac causes of chest pain in children and adolescents, emphasizing the critical role of clinical history and physical examination. In most cases, these alone can help determine the underlying cause of chest pain. Based on the suspected etiology, targeted investigations are recommended for patients presenting with chest pain in the emergency department. This figure also provides guidance on when referral to a pediatric cardiologist is warranted.

Medical treatment for chest pain in children primarily involves addressing the underlying cause of the pain, which can range from benign conditions like precordial catch syndrome to more serious causes such as cardiac or pulmonary issues. When the pain is due to benign causes like musculoskeletal discomfort, analgesics such as nonsteroidal anti-inflammatory drugs (NSAIDs) like ibuprofen or acetaminophen are commonly prescribed to alleviate discomfort [46]. In cases where anxiety or stress is a significant factor, which is often the case in chest pain of non-cardiac origin, anxiolytics may be indicated [47]. These medications, such as benzodiazepines, can help reduce anxiety, especially if the child exhibits heightened concern or panic related to their symptoms. However, the use of anxiolytics is typically reserved for children with persistent or severe anxiety that exacerbates their chest pain, and they are usually prescribed after careful assessment by a healthcare professional to rule out other causes. Additionally, cognitive-behavioral therapy (CBT) and reassurance from healthcare providers play an essential role in managing anxiety-related chest pain, reducing the need for pharmacological intervention [47]. Ensuring an accurate diagnosis through history and physical examination is crucial to avoid unnecessary testing and treatment for non-cardiac chest pain, which is common in pediatric patients.

## 5. Conclusions

Chest pain is a frequent cause of concern for parents, often leading to emergency department visits. Unlike in adults, where chest pain is more likely to indicate a serious cardiovascular issue, pediatric chest pain is typically benign. The most common causes of chest pain in children are musculoskeletal conditions, particularly precordial catch syndrome, followed by anxiety-related causes. Infections, while possible, are less frequent. Dysautonomia and cardiac neuropathy are uncommon causes but should be considered when there are other clinical signs of autonomic dysfunction. For clinicians, the challenge lies in ruling out serious uncommon cardiac issues.

Our review highlights several key new findings that differ from previous literature reviews. Notably, recent studies indicate that while the prevalence of serious cardiac causes remains low (less than 1%), there is increasing recognition of conditions such as myocarditis and pericarditis linked to viral infections, including SARS-CoV-2 and its vaccines. These insights were not as emphasized in prior reviews, particularly concerning the impact of COVID-19 on pediatric chest pain diagnoses. Additionally, this review underscores that diagnostic approaches to pediatric chest pain vary widely, with a lack of consensus on standard evaluation algorithms. The emerging data support the need for more structured algorithms to minimize unnecessary diagnostic testing while ensuring that serious conditions are not missed. This contrasts with earlier reviews, which largely focused on identifying individual causes rather than improving the overall clinical approach.

From a clinical perspective, this article also elaborates on the practical management of pediatric chest pain, emphasizing the need for clinicians to carefully evaluate the condition based on the patient’s clinical history and “red flag” symptoms. While past research has focused predominantly on non-cardiac causes, this review calls attention to the delicate balance between identifying potentially serious cases and avoiding unnecessary investigations. Clinicians are advised to adopt a cautious, evidence-based approach, while diagnostic tests should be used judiciously to reduce healthcare resource burden. The findings presented herein offer a more clinically focused framework for the management of pediatric chest pain, providing valuable guidance for physicians in emergency and clinical settings.

While many of the causes of chest pain in children and adolescents are not dangerous, the potential risk of overlooking a serious condition, particularly a cardiac one, remains a significant concern for both practitioners and parents. Therefore, in line with the SCAMPs guidelines, all children presenting with chest pain should undergo a thorough evaluation by a pediatric clinician. This evaluation should include a detailed history and physical examination, with particular attention to red flag symptoms. If any abnormalities are detected during the physical examination, or if there is a personal or family history of cardiac disease (such as specific arrhythmias, congenital heart defects, or sudden death), the child should be referred to a pediatric cardiologist.

It is crucial to carefully consider which diagnostic tests are necessary for children presenting with chest pain. The goal is to identify those at risk of life-threatening conditions while avoiding unnecessary tests that could lead to an inappropriate use of healthcare resources and additional stress for the child and their family. Routine tests may include an ECG and chest X-ray. In a small subset of cases, blood tests (such as troponin levels) or echocardiography performed by a specialized pediatric cardiologist may be indicated.

In conclusion, the evidence strongly suggests that children and adolescents presenting to an emergency department with chest pain should be initially evaluated by general pediatricians, with only a select group requiring further evaluation by a pediatric cardiologist or additional diagnostic testing. However, there is still considerable variability in the diagnostic approach taken by pediatricians, often influenced by the thoroughness of the initial evaluation and history-taking. Future studies should focus on the development and implementation of standardized algorithms for the evaluation of pediatric chest pain. Such algorithms could help streamline the diagnostic process, reducing the number of unnecessary tests while ensuring that serious conditions are not missed. Research should aim to assess the impact of these algorithms on clinical outcomes, resource utilization, and patient and family satisfaction. By establishing a more uniform approach to pediatric chest pain, it may be possible to improve care quality, reduce healthcare costs, and alleviate the anxiety associated with this common symptom.

## Figures and Tables

**Figure 1 jcm-13-06659-f001:**
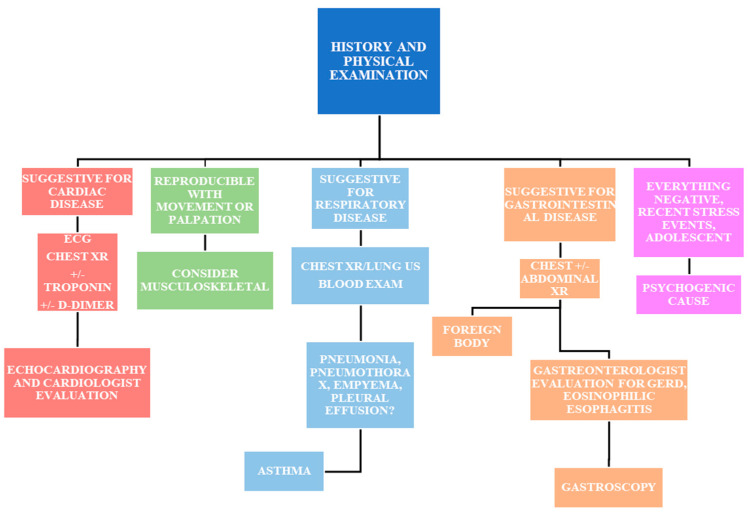
Management of children and adolescents presenting with chest pain in an emergency department.

**Table 1 jcm-13-06659-t001:** Red flags in pediatric chest pain: symptoms, underlying conditions, diagnostic steps, and therapeutic actions.

Symptoms	Potential Underlying Diseases	Incidence Data	Initial Diagnostic Steps	Urgent Therapeutic Actions/Hospital Admission Criteria
Exertional chest pain, syncope	Cardiomyopathy (e.g., hypertrophic), congenital coronary anomalies, arrhythmias	Rare, but higher risk in patients with family history of cardiac disease or sudden death	ECG, echocardiography	Immediate referral to pediatric cardiology; hospital admission for monitoring
Chest pain radiating to the arm/back	Acute myocardial inflammation (myocarditis, pericarditis)	Myocarditis is rare (<1% of pediatric chest pain cases); pericarditis frequency increased post-viral infections	ECG, troponin, echocardiography	Hospital admission, supportive care, anti-inflammatory therapy if indicated
Dizziness, palpitations	Arrhythmias, including long QT syndrome	Rare in general population but associated with genetic predisposition	ECG (look for QT interval abnormalities), Holter monitoring	Antiarrhythmics if required, hospital admission if arrhythmia confirmed
Sudden-onset chest pain with dyspnea	Pulmonary embolism, pneumothorax, pneumomediastinum	Pulmonary embolism rare in children (<1%), pneumothorax more common in adolescents	Chest X-ray, lung ultrasound, CT for suspected embolism	Oxygen therapy; hospital admission for pneumothorax or embolism
Persistent/recurrent vomiting	Gastroesophageal reflux disease (GERD), esophagitis	GERD common in children with non-cardiac chest pain (~5–8%)	Upper GI series, pH probe testing	Antacids, proton pump inhibitors; hospital admission for severe esophagitis
Cough, fever, difficulty breathing	Pneumonia, pleuritis, bronchitis	Pneumonia frequency ~3–9% in pediatric chest pain presentations	Chest X-ray, complete blood count (CBC)	Antibiotics if bacterial; consider admission if severe respiratory distress
Associated abdominal pain	Gastrointestinal disorders (GERD, esophageal foreign body, gastritis)	Foreign body ingestion <2% of chest pain cases in children	Abdominal X-ray, endoscopy (if foreign body suspected)	Remove foreign body if confirmed; antacids for GERD
New or worsening heart murmur	Valvular disease, congenital heart defect	Rare; typically identified earlier but may present in adolescence	ECG, echocardiography	Urgent cardiology referral, hospital admission if symptomatic
Severe, sharp, unilateral chest pain	Musculoskeletal causes (e.g., precordial catch syndrome, costochondritis)	Precordial catch syndrome common, particularly in adolescents	Physical examination; consider X-ray to exclude trauma	Analgesics; rarely requires hospital admission
Profuse sweating, pallor	Myocardial ischemia (rare in children), severe arrhythmias	Extremely rare in children, but can indicate severe cardiac pathology	ECG, echocardiography, blood pressure measurement	Immediate hospitalization, oxygen therapy, ECG monitoring

## Data Availability

Not applicable.

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
