# Peer review of "Approaches to Pediatric Chest Pain: A Narrative Review"

_jcm, 2024, doi:10.3390/jcm13226659_

Round 1

Reviewer 1 Report

Comments and Suggestions for Authors

Thank you for submitting your review article to JCM.

This article focuses on chest pain, which is a common chief complaint of children visiting hospitals, health care facilities, and emergency department. The overall content is about approaches to pediatric chest pain and is in the form of a scoping review. Please refer to the peer review evaluation of this paper below. Please also refer to the comments from the other reviewers. I look forward to your diligent responses.

Major

1. Although this review describes a comprehensive medical approach to chest pain in children, the overall impression is that it is dominated by a basic primary diagnostic medical approach. What are the new findings for chest pain in children that differ from previously published reviews? Clarifying this point will enhance the value of the review article. Conversely, if the new message point is not clear, it is a list of past papers.

2. This article describes a basic approach to chest pain in children. However, it is a submission to a Clinical Medicine journal. So, it would be better to arrange it into a review article with more clinical information.

Minor

1.L50-:  You have focused on chest pain and myocarditis after COVID-19 coronavirus vaccine, but what about influenza vaccine and smallpox vaccine? Is it necessary to mention in this paper that chest pain, although rare, is present after various vaccines?

2.L66-: In the clinical medicine of chest pain, the time of onset of chest pain is important. It may have occurred in the morning, at a time before bed, or others. The circumstances of onset of chest pain are also important. It may have occurred after using the restroom or during exercise.

3.  L87-: I agree that a detailed physical examination is important as an approach to chest pain. In this case, it is important to observe the body shape. In childhood, it is important to focus on growth disturbances, such as whether the patient is short or tall, underweight or obese. Also, the presence or absence of scoliosis and a variety of bone diseases should be noted.

4. L87-: Is it necessary to describe auscultatory findings in the chest related to chest pain?

5. L96-: Which should be done first, 2.2.1 electrocardiogram (ECG) or 2.2.2 chest X-ray? Please also consider the order of the chapters.

6. There is no text on chest pain in pediatric febrile patients. It is important to note enterovirus and coxsackie virus as prevalent viral infections, as well as Bornholm's disease, which is a viral myocarditis.

7. Does a child with chest pain need general blood tests, such as WBC, CRP, LDH, AST, CPK, etc.?

8. The troponin test for children includes TnT, which reflects tropomyosin, and TnI, which looks at actomyosin. Is it necessary to examine both separately?

9. The specific difference in the heart-to-thorax ratio (CTR) at different ages is also important.

10. It should be noted that there is a wide range of vital signs such as blood pressure, pulse rate, and respiratory rate that vary with age.

11. Regarding cardiac MRI, when should it be done in children with chest pain?

12. What is the age of the children you consider to be the subject of this paper? Will neonates and infants be excluded? As for adolescent children, would they be up to 12, 15, or 18 years of age?

13. I understood the message that many chest pains are benign. So what are the most common causes of chest pain? Is it precordial catch syndrome? Is it an infection such as mycoplasma? Or is it chest pain due to dysautonomia, cardiac neuropathy, or anxiety? This is a point of great concern for many clinicians.

My comments as a reviewer pointed out many of the concerns that pediatricians have when they see a child with chest pain in the emergency room. I hope that this review will be revised into a clinically useful and important review of pediatric chest pain.

Best regards,

Dr. Reviewer

Author Response

Thank you for submitting your review article to JCM.

This article focuses on chest pain, which is a common chief complaint of children visiting hospitals, health care facilities, and emergency department. The overall content is about approaches to pediatric chest pain and is in the form of a scoping review. Please refer to the peer review evaluation of this paper below. Please also refer to the comments from the other reviewers. I look forward to your diligent responses.

Re: Thank you for your comments. We revised the manuscript according to your suggestions.

Major

  1. Although this review describes a comprehensive medical approach to chest pain in children, the overall impression is that it is dominated by a basic primary diagnostic medical approach. What are the new findings for chest pain in children that differ from previously published reviews? Clarifying this point will enhance the value of the review article. Conversely, if the new message point is not clear, it is a list of past papers.

Re: A section on new findings for chest pain in children that differ from previously published reviews has been added in the Conclusions (p. 10).

  1. This article describes a basic approach to chest pain in children. However, it is a submission to a Clinical Medicine journal. So, it would be better to arrange it into a review article with more clinical information.

Re: Further clinical information has been added throughout the manuscript, including the Conclusions (pp. 1-10).

Minor

1.L50-:  You have focused on chest pain and myocarditis after COVID-19 coronavirus vaccine, but what about influenza vaccine and smallpox vaccine? Is it necessary to mention in this paper that chest pain, although rare, is present after various vaccines?

Re: Added (p. 2).

2.L66-: In the clinical medicine of chest pain, the time of onset of chest pain is important. It may have occurred in the morning, at a time before bed, or others. The circumstances of onset of chest pain are also important. It may have occurred after using the restroom or during exercise.

Re: Added (p. 2).

  1. L87-: I agree that a detailed physical examination is important as an approach to chest pain. In this case, it is important to observe the body shape. In childhood, it is important to focus on growth disturbances, such as whether the patient is short or tall, underweight or obese. Also, the presence or absence of scoliosis and a variety of bone diseases should be noted.

Re: Added (p. 3).

  1. L87-: Is it necessary to describe auscultatory findings in the chest related to chest pain?

Re: We improved this section and we think that the sentence on auscultary findings can remain has written (p. 3).

  1. L96-: Which should be done first, 2.2.1 electrocardiogram (ECG) or 2.2.2 chest X-ray? Please also consider the order of the chapters.

Re: In the ER, ECG is usually performed by the nurses at admission for chiest pain, whereas chest X-ray is requested by the MDs in a part of the cases after medical visits.  

  1. There is no text on chest pain in pediatric febrile patients. It is important to note enterovirus and coxsackie virus as prevalent viral infections, as well as Bornholm's disease, which is a viral myocarditis.

Re: Added (p. 3).

  1. Does a child with chest pain need general blood tests, such as WBC, CRP, LDH, AST, CPK, etc.?

Re: Clarified (p. 4).

  1. The troponin test for children includes TnT, which reflects tropomyosin, and TnI, which looks at actomyosin. Is it necessary to examine both separately?

Re: Clarified (p. 4).

  1. The specific difference in the heart-to-thorax ratio (CTR) at different ages is also important.

Re: Added (p. 6).

  1. It should be noted that there is a wide range of vital signs such as blood pressure, pulse rate, and respiratory rate that vary with age.

Re: Added (p. 3).

  1. Regarding cardiac MRI, when should it be done in children with chest pain?

Re: Added.

  1. What is the age of the children you consider to be the subject of this paper? Will neonates and infants be excluded? As for adolescent children, would they be up to 12, 15, or 18 years of age?

Re: Clarified (p. 2).

  1. I understood the message that many chest pains are benign. So what are the most common causes of chest pain? Is it precordial catch syndrome? Is it an infection such as mycoplasma? Or is it chest pain due to dysautonomia, cardiac neuropathy, or anxiety? This is a point of great concern for many clinicians.

Re: This issue has been considered in the Conclusions (p. 11).

My comments as a reviewer pointed out many of the concerns that pediatricians have when they see a child with chest pain in the emergency room. I hope that this review will be revised into a clinically useful and important review of pediatric chest pain.

Re: Thank you for your suggestions. We improved the manuscript according to your comments and those of the other reviewers. We hope that you could accept it in the revised form.

Best regards,

Dr. Reviewer

Reviewer 2 Report

Comments and Suggestions for Authors

This is a well written review article on the evaluation of pediatric chest pain. It encompasses a thorough literature review as well as some of the current clinical testing/management strategies.

While I'm not sure it adds much to the current literature it is certainly a thorough review of the subject. It reads very easily and is constructed well. The review provides a nice suggested framework/clinical approach to patients with examples of red flags. 

Author Response

This is a well written review article on the evaluation of pediatric chest pain. It encompasses a thorough literature review as well as some of the current clinical testing/management strategies.

Re: Thank you very much for the appreciation of our manuscript. We revised the manuscript according to your comments and those of the other reviewers.

While I'm not sure it adds much to the current literature it is certainly a thorough review of the subject. It reads very easily and is constructed well. The review provides a nice suggested framework/clinical approach to patients with examples of red flags. 

Re. A section on new findings for chest pain in children that differ from previously published reviews has been added in the Conclusions (p. 10).

Reviewer 3 Report

Comments and Suggestions for Authors

Review

Approach to Pediatric Chest Pain: a Scoping Review

Comments:

The article is well written and informative. Although it does not provide many innovative insights, it addresses an important topic for clinical pediatricians and pediatric cardiologists, as chest pain in children and adolescents is often a diagnostic challenge. It should therefore be of interest for the Journal’s readership. Nevertheless, in my opinion, some missing points need to be addressed.

Please consider the following comments and supplements:

-          The numbering in Table 1 is incomplete (“1. Cardiac Symptoms… 2. Respiratory Symptoms”).

-          Section 2.1.: Important information in clinical history that is not mentioned in the text is potential thrombophilia or previous thrombosis, as well as smoking and hormonal contraception in female patients. Preceding febrile infections can be in particular an indication for myocarditis. Sports activities, possibly limited physical performance, unilateral load, as well new or unusual strain must be inquired about.

-          Section 2.2.: It is essential to assess the patient’s blood pressure, oxygen saturation and heartrate.

-          Section 2.2.4.: It would be informative to comment on the significance of other cardiac enzymes, which are very often (unnecessarily?) measured in children with chest pain, e.g. creatine kinase and natriuretic peptide.

-          Lines 184 – 190: Although rare, it would be important to mention not only anomalous coronary origin but also congenital anomalies in the course of coronary arteries (e.g. intramural or interarterial) that can cause chest pain during exercise.

-          Aortic dissection is also a rare but serious cause of severe chest pain usually radiating to the back, which must be considered in particular in patients with connective tissue diseases (e.g. Marfan...). It is very important to consider this diagnosis, as it is an absolute emergency that must have immediate diagnostic and therapeutic consequences!

-          Section 3.2.: Pleuritis, particularly epidemic pleurodynia, causes severe chest pain and should therefore be stated.

-          Lines 226 - 234: Esophageal rupture is one of the most common causes for pneumomediastinum and should therefore be mentioned in this section.

General remarks:

-          Age-specific symptoms and causes should be given greater consideration, as some causes are more likely in certain age groups. For example, young children in particular may express “chest pain” with tachycardia, but also with actual abdominal pain, which is why a full body examination is necessary!

-          It is important to state, that chest pain is generally not a common symptom of congenital heart disease. Exceptions to this are left ventricular outflow tract obstructions such as aortic valve stenosis or coronary anomalies.

Comments on the Quality of English Language

The article is well written and structured.

Author Response

Approach to Pediatric Chest Pain: a Scoping Review

Comments:

The article is well written and informative. Although it does not provide many innovative insights, it addresses an important topic for clinical pediatricians and pediatric cardiologists, as chest pain in children and adolescents is often a diagnostic challenge. It should therefore be of interest for the Journal’s readership. Nevertheless, in my opinion, some missing points need to be addressed.

Re: Thank you for your comments. We revised the manuscript according to your recommendations.

Please consider the following comments and supplements:

-          The numbering in Table 1 is incomplete (“1. Cardiac Symptoms… 2. Respiratory Symptoms”).

Re: We removed the numbers from Table 1 because they are not needed.

-          Section 2.1.: Important information in clinical history that is not mentioned in the text is potential thrombophilia or previous thrombosis, as well as smoking and hormonal contraception in female patients. Preceding febrile infections can be in particular an indication for myocarditis. Sports activities, possibly limited physical performance, unilateral load, as well new or unusual strain must be inquired about.

Re: This information has been added (p. 2).

-          Section 2.2.: It is essential to assess the patient’s blood pressure, oxygen saturation and heartrate.

Re: Added (p. 3).

-          Section 2.2.4.: It would be informative to comment on the significance of other cardiac enzymes, which are very often (unnecessarily?) measured in children with chest pain, e.g. creatine kinase and natriuretic peptide.

Re: Added (p. 4).

-          Lines 184 – 190: Although rare, it would be important to mention not only anomalous coronary origin but also congenital anomalies in the course of coronary arteries (e.g. intramural or interarterial) that can cause chest pain during exercise.  Aortic dissection is also a rare but serious cause of severe chest pain usually radiating to the back, which must be considered in particular in patients with connective tissue diseases (e.g. Marfan...). It is very important to consider this diagnosis, as it is an absolute emergency that must have immediate diagnostic and therapeutic consequences!

Re: These rare causes of chest pain have been added (p. 7).

-          Section 3.2.: Pleuritis, particularly epidemic pleurodynia, causes severe chest pain and should therefore be stated.

Re: Added (p. 8).

-          Lines 226 - 234: Esophageal rupture is one of the most common causes for pneumomediastinum and should therefore be mentioned in this section.

Re: Added (p. 8).

General remarks:

-          Age-specific symptoms and causes should be given greater consideration, as some causes are more likely in certain age groups. For example, young children in particular may express “chest pain” with tachycardia, but also with actual abdominal pain, which is why a full body examination is necessary!

Re: This remark has been included in the text (p. 3).

-          It is important to state, that chest pain is generally not a common symptom of congenital heart disease. Exceptions to this are left ventricular outflow tract obstructions such as aortic valve stenosis or coronary anomalies.

Re: Added (p. 8).

Round 2

Reviewer 1 Report

Comments and Suggestions for Authors

Dear Author,

Thank you again for your submission to JCM.

The authors have responded sincerely to many peer review comments. The text has been fully revised.

Again, in the opinion of the reviewers, would you please consider the following two points as follows;

Minor 1.

You may want to add a literature review and further elaborate on the precordial catch syndrome.

Minor 2.

It would be useful clinical information if you could organize and describe the medical treatments for chest pain in children. In doing so, please include indications for anxiolytics as well as analgesics.

This review contains very useful information, and I look forward to its publication in JCM. I look forward to its publication in JCM .

(Please note that the actual publication is at the discretion of the editor.)

Best regards,

Dr. Reviewer

Author Response

Dear Author,

 Thank you again for your submission to JCM.

The authors have responded sincerely to many peer review comments. The text has been fully revised.

Again, in the opinion of the reviewers, would you please consider the following two points as follows;

Re: Thank you very much for your positive evaluation. We improved the text according to your suggestions and we hope that it could be considered acceptable for publication in its present form. 

Minor 1.

You may want to add a literature review and further elaborate on the precordial catch syndrome.

 Re: We added a paragraph as suggested (p. 10) with four new references (p. 17).

Minor 2.

It would be useful clinical information if you could organize and describe the medical treatments for chest pain in children. In doing so, please include indications for anxiolytics as well as analgesics.

Re: We added a paragraph as suggested (p. 12) with two new references (p. 18). 

This review contains very useful information, and I look forward to its publication in JCM. I look forward to its publication in JCM .

Re: Thank you very much for your suggestions. We further improved the text as recommended and we hope that it could be considered suitable for publication in its present form.